# Use of Paclitaxel to Successfully Treat Children, Adolescents, and Young Adults with Kaposi Sarcoma in Southwestern Tanzania

**DOI:** 10.3390/children8040275

**Published:** 2021-04-02

**Authors:** Hamidu Adinani, Liane Campbell, Nader Kim El-Mallawany, Jeremy Slone, Parth Mehta, Jason Bacha

**Affiliations:** 1Department of Health and Social Welfare, Tanzania Ministry of Health, Community De-velopment, Gender, Elderly and Children, Tarime District, Mara Region 31401, Tanzania; hamidu5766@gmail.com; 2Department of Pediatrics, Baylor College of Medicine Children’s Foundation—Tanzania, Mbeya 53107, Tanzania; bacha@bcm.edu; 3Baylor College of Medicine International Pediatric AIDS Initiative, Texas Children’s Hospital, Houston, TX 77030, USA; 4Department of Pediatrics, Baylor College of Medicine, Houston, TX 77030, USA; Nader.El-Mallawany@bcm.edu (N.K.E.-M.); jsslone@texaschildrens.org (J.S.); psmehta@texaschildrens.org (P.M.); 5Global HOPE, Texas Children’s Cancer and Hematology Centers, Texas Children’s Hospital, Houston, TX 77030, USA

**Keywords:** Kaposi sarcoma, HIV/AIDS, pediatrics, global health, chemotherapy, HIV-related malignancy, life-limiting

## Abstract

Treating Kaposi sarcoma (KS) in children, adolescents, and young adults (AYA) remains a challenge in low- and middle-income countries (LMIC) where chemotherapy options and availability are limited. We describe a retrospective cohort review of pediatric patients with KS treated with paclitaxel in Mbeya, Tanzania, between 1 March 2011 and 31 December 2019. Paclitaxel was given to patients who had KS relapse, a contraindication to bleomycin, vincristine, and doxorubicin (ABV), special circumstances in which a clinician determined that paclitaxel was preferable to ABV, or experienced treatment failure, defined as persistent KS symptoms at the completion of treatment. All patients also received multidisciplinary palliative care. Seventeen patients aged 5.1–21.3 years received paclitaxel, of whom 47.1% (8/17) had treatment failure, 29.4% (5/17) received paclitaxel as initial treatment, and 23.5% (4/17) had relapsed. All HIV positive patients (16/17) were given anti-retroviral therapy (ART) and 87.5% (14/16) achieved viral load <1000 cp/mL. At censure, 82.3% (14/17) of patients were alive—71.4% (10/14) achieved complete clinical remission and 28.6% (4/14) achieved a partial response. The median follow up was 37.3 months (range 8.0–83.5, IQR 19.7–41.6), and no patients were lost to follow up. In this cohort, high rates of long-term survival and favorable outcomes were possible with paclitaxel treatment.

## 1. Introduction

Driven by the HIV/AIDS pandemic, Kaposi sarcoma (KS) is one of the most common pediatric malignancies in sub-Saharan African settings with endemicity of Human gammaherpesvirus 8 (HHV-8), the causative organism for KS, and uncontrolled HIV/AIDS among children and adolescents [1,2,3,4,5,6]. Despite improvements in the availability of anti-retroviral therapy (ART) in sub-Saharan Africa, KS remains an important malignancy, and in Tanzania, the prevalence of KS among HIV positive (HIV+) adult patients receiving HIV care was reported to be 4.6% [7]. While many adult patients with KS meet criteria for WHO severe immunosuppression, KS has also been observed in patients on ART who have achieved immune reconstitution and viral control [8,9].

While there is growing attention to the diagnosis and treatment of KS in adults, knowledge of KS in children and adolescents remains limited. KS is known to have a distinct clinical presentation in children and adolescents relative to adults, with lymphadenopathy more commonly seen in children, in addition to the skin lesions and visceral involvement that can define the presentation of adults with KS [9,10]. Existing data has shown that favorable clinical outcomes are achievable in pediatric and adolescent KS patients utilizing a risk stratified approach to chemotherapy [11]. In a recent cohort of pediatric and adolescent patients with KS from Malawi, patients with mild and moderate cutaneous and oral disease and those with lymphadenopathic predominant KS were able to achieve relatively high overall and event free survival with bleomycin and vincristine (BV) treatment. Patients with woody edema also achieved relatively high overall survival but experienced more events. However, patients with visceral disease had lower overall and event free survival, despite receiving treatment with BV [11].

While no consensus guidelines exist on the best treatment options in low- and middle-income countries (LMICs) for children and adolescents with difficult to treat KS, lessons can be learned from the adult KS experience with paclitaxel. In a Malawian cohort of adults with HIV associated KS, paclitaxel was utilized to effectively treat patients with severe KS disease and patients who failed to respond to BV chemotherapy [12]. Paclitaxel has also been utilized to treat adults with relapsed visceral KS [13] and aggressive refractory classic KS [14]. In a large multi-country randomized control trial of adults with HIV associated KS, patients treated with ART and paclitaxel had better clinical outcomes relative to those treated with oral etoposide, or vincristine and bleomycin [15].

There is limited published information regarding treatment of pediatric patients with KS with paclitaxel, but in a cohort of pediatric patients with KS from Mozambique who were treated with monthly paclitaxel and ART, 67.9% (19/28) achieved long term remission, including patients with T1 disease [16]. Similarly, in a case series from Botswana, paclitaxel was used successfully to treat two pediatric patients with relapsed visceral KS [17]. Additional evidence on the use of paclitaxel for pediatric, and adolescents and young adults (AYA) patients with KS is desperately needed to optimize outcomes of these patients. To address this gap, we describe the clinical characteristics and outcomes of pediatric and AYA patients with KS who were treated with paclitaxel in Mbeya, Tanzania.

## 2. Materials and Methods

A retrospective cohort analysis was performed of patients diagnosed with KS and treated with paclitaxel at the Baylor Tanzania Children’s Foundation Mbeya Centre of Excellence (COE) between 1 March 2011 and 31 December 2019.

### 2.1. KS Diagnosis

Patients who presented with prototypical violaceous skin and/or oral lesions; firm, fixed lymphadenopathy greater than 2 cm in multiple sites; or woody edema were clinically diagnosed with KS. Histopathologic confirmation was not routinely available in this setting due to a variety of logistical and laboratory limitations. However, clinical diagnosis was confirmed with histopathology in difficult cases where the diagnosis was unclear. Patients with dysphagia or persistent bloody stool unresponsive to anti-microbials and without alternative diagnoses were diagnosed with gastrointestinal KS disease; those with pleural effusions unresponsive to anti-tuberculosis therapy or a reticulonodular pattern on chest X-ray were diagnosed with pulmonary KS. At diagnosis, patients underwent a detailed clinical exam and were staged utilizing the modified Lilongwe pediatric KS staging classification [18]. Patients with fewer than 10 skin lesions or a single flat palate lesion were considered to have mild disease (stage 1A). Patients with moderate disease included those with 10–20 skin lesions (stage 1B), those with lymphadenopathy as their predominant symptom (stage 2), and those with woody edema (stage 3). Patients with more than 20 skin lesions and/or clinical or radiographic evidence of visceral disease were defined as having severe disease (stage 4) disease. The modified Lilongwe pediatric KS staging classification was devised based on a retrospective review of clinical features associated with outcomes among a cohort of pediatric and adolescent patients with HIV associated KS [11]. AIDS Clinical Trial Group (ACTG) TIS (T: tumor; I: immune system; S: systemic illness) staging in which 0 was representative of low risk and 1 was representative of high risk was performed retrospectively to provide additional descriptive information about the cohort. Patients were defined as having I1 disease if they had CD4 counts less than 200 cells/µL [19] or less than 15% in the case of children under the age of 5 years [16]. An important distinction between the modified Lilongwe pediatric KS staging classification and ACTG TIS staging is the classification of patients with >20 skin lesions. These patients would receive a T0 designation in the TIS staging classification. However, because retrospective analysis of these patients have demonstrated poor outcomes when treated with BV chemotherapy, in the modified Lilongwe pediatric KS staging classification, patients with >20 skin lesions are classified as having severe disease (stage 4) and are treated with ABV chemotherapy [11,20].

### 2.2. Approach to Chemotherapy

The modified Lilongwe pediatric Kaposi sarcoma staging classification was utilized to guide treatment approach (Figure 1). Patients with mild to moderate cutaneous or oral disease, lymphadenopathic predominant disease, or woody edema received initial treatment with BV; those who did not achieve complete clinical remission (CCR) or demonstrate a partial response (PR) after initial treatment with BV received intensified chemotherapy with bleomycin, vincristine, and doxorubicin (ABV). Patients with woody edema with persistent symptoms that adversely impacted their quality of life, for example, causing difficulty in ambulation, received intensified therapy with ABV. Patients diagnosed with visceral disease received initial treatment with ABV.

Treatment response was monitored clinically—CCR was defined as sustained and complete disappearance of all KS lesions and/or full resolution of clinical symptoms and/or CXR findings in the case of visceral KS. PR was defined as at least 50% reduction in evaluable lesions of the skin, oral cavity, or viscera, with the recognition that establishing a reduction in gastrointestinal or pulmonary lesions may be difficult or impossible with the available diagnostic capabilities [22]. In the case of patients with woody edema, PR was defined as a significant improvement in quality of life [11]. Relapse was defined as a recurrence or worsening of any KS clinical features after an initial response to treatment, and/or the appearance of new KS lesions. For patients with woody edema, treatment failure was defined as persistent symptoms without improvement in quality of life at the end of initial treatment; for patients with visceral/disseminated disease, treatment failure was defined as persistent disease at the end of initial treatment. Cumulative lifetime doses of bleomycin and doxorubicin did not exceed 270 U/m^2^ and 300 mg/m^2^, respectively.

Paclitaxel was selected for patients who experienced treatment failure or developed KS relapse. Paclitaxel was given as initial treatment in special circumstances to patients who had a contraindication to ABV at KS diagnosis, for example, an allergic reaction to BV or ABV, or pre-existing cardiomyopathy, and in cases of clinician preference. Paclitaxel was given at 100–135 mg/m^2^ every 3–4 weeks for 6 cycles with pre-medications of dexamethasone 10 mg/m^2^ and an H-1, and an H-2 receptor antagonist. Infusions were given over a minimum of 4 hours with regular vital sign and toxicity monitoring. All chemotherapy was given in an outpatient setting. Patients who did not achieve CCR or PR after 6 cycles were given up to an additional 6 cycles of paclitaxel, depending on the treatment goals of the patient and their family. All treatment was free of charge to patients and families including costs of chemotherapy and diagnostics. Program costs were covered by private donations and a True Colours Trust Small Grants for Africa administered by the African Palliative Care Association which covered some costs of chemotherapy and social support.

### 2.3. HIV Care

All HIV positive patients were treated with ART according to Tanzanian national guidelines [23]. Patients who were being treated with a non-nucleoside reverse transcriptase inhibitor (NNRTI)-based regimen and who were diagnosed with HIV treatment failure were switched to a protease inhibitor (PI)-based regimen. Integrase strand inhibitors were not available in Tanzania during the study period. All patients received cotrimoxazole prophylaxis during and for a minimum of 6 months after completing chemotherapy. Immunosuppression was categorized according to World Health Organization age based standards in which severe immunosuppression corresponded to CD4 counts of less than 200 cells/µL for patients over the age of 5 years, less than 15% for patients between the ages of 13 and 59 months or less than 20% for patients between the ages of 0 and 12 months. Advanced immunosuppression was defined as CD4 counts between 200 and 349 cells/µL for patients over the age of 5 years, between 15 and 19% for patients between the ages of 13 and 59 months or between 20 and 24% for patients between the ages of 0 and 12 months. Mild immunosuppression was defined as CD4 counts between 350 and 499 cells/µL for patients over the age of 5 years, between 20 and 24% for patients between the ages of 13 and 59 months or between 25 and 34% for patients between the ages of 0 and 12 months [24]. KS immune reconstitution inflammatory syndrome (IRIS) was defined as development of or clinical worsening of features of KS in the 6 months after initiating ART in a patient not receiving chemotherapy [20].

### 2.4. Nutritional Assessment and Support

All patients underwent nutritional analysis at time of KS diagnosis and at each subsequent clinic visit utilizing weight for height z scores, body mass index (BMI) and mid upper arm circumference (MUAC). For children between 6 and 60 months, severe acute malnutrition (SAM) was defined a weight/height z score worse than −3 standard deviations (SD) from the median, the presence of nutritional edema, or MUAC < 11.5 cm. Moderate acute malnutrition (MAM) was defined as weight/height z score between −2 and −3 SD or MUAC between 11.5 cm and 12.5 cm. For patients over 60 months of age, SAM was defined as BMI z score worse than −3 SD for age and gender, the presence of nutritional edema or MUAC < 13.5 cm for those between the ages of 5 to 9 years and <16 cm for those between the ages of 10 and 14 years. Moderate acute malnutrition in those >60 months was defined as those with BMI z score between −3 and −2 SD or MUAC between 13.5 and 14.5 cm for those between the ages of 5 and 9 years or between 16 and 18.5 cm for those between the ages of 10 and 14 years [25,26,27]. Patients who had moderate or severe malnutrition received nutritional supplements such as ready to use therapeutic foods or formula milks. Severely malnourished patients also received broad spectrum antibiotics, micronutrient supplementation and deworming in accordance with WHO recommendations for management of severe acute malnutrition [25].

### 2.5. Palliative Care

All patients diagnosed with KS were enrolled in the COE’s multidisciplinary palliative care program and received psychosocial support, transport assistance, phone follow up support, and pain and symptom management. Patients had the opportunity to participate in a wish making activity and were given access to the COE’s 24-h palliative care nurse hotline.

### 2.6. Statistical Analysis

Demographic and clinical information was extracted from the medical record. Chi-square and Fisher’s Exact Probability tests were used to determine the relationship between participants’ characteristics and outcomes. A *p*-value of <0.05 was considered statistically significant.

### 2.7. Ethical Clearance

Ethical approval was obtained from the Mbeya Medical Research and Ethics Committee and the National Institute of Medical Research (NIMR) in Tanzania, and the Institutional Review Board, Baylor College of Medicine, Houston, TX, USA. Waiver of consent was approved by all committees as this retrospective study analyzed de-identified data.

## 3. Results

### 3.1. Clinical Characteristics

During the study period, there were 71 total patients treated for KS, of whom 17 (23.9%) were treated with paclitaxel and included in the analysis. Patients in this cohort had a median age of 13.0 years (IQR 9.3–15.8) and 12 patients were adolescents or young adults (AYA) (Table 1). T1 disease was seen in 12 patients, 8 had visceral and/or disseminated skin/oral KS (Lilongwe stage 4 disease), and 3 had histopathologic confirmation of their clinical disease. Severe acute malnutrition was present in 6. Among HIV+ patients (16), 10 met criteria for WHO severe immunosuppression. Four patients presented with KS IRIS, all of whom developed symptoms of KS shortly after initiation of ART (range of 0.8–4.2 months).

The 54 patients who were not treated with paclitaxel during this study period included 5 patients with Lilongwe stage 1 disease (2 patients were 1A and 3 patients were 1B), 17 patients with Lilongwe stage 2 disease, 17 patients with Lilongwe stage 3 disease, and 15 patients with Lilongwe stage 4 disease. Of these patients, 53 were HIV positive and 1 was HIV negative. At time of KS diagnosis, of the 53 patients who were HIV positive, 39 patients were receiving ART and 14 patients had not yet been initiated on ART. Of the 54 patients who were not treated with paclitaxel, 33 were treated with BV, 17 required intensified treatment with ABV, and 4 died prior to initiation of chemotherapy. No patients were treated with ART alone. Of these patients, 38 survived and 16 died (Table 2).

### 3.2. Outcomes

Among the KS patients who were treated with paclitaxel, eight had experienced treatment failure, five received paclitaxel as initial treatment for KS, and four had relapsed KS. Patients in the cohort had a median follow up time of 37.3 months (range 8.0–83.5, IQR 19.7–41.6), during which 14 patients were alive. No patients were lost to follow up. Among living patients (14), 10 achieved CCR, 4 achieved a PR. No patients experienced allergic reactions including anaphylaxis during or after paclitaxel infusions, one patient developed grade 3 neutropenia but did not have associated fever or sepsis, and one patient required a blood transfusion for anemia. The patients who presented with KS IRIS did not require additional specific treatment IRIS other than chemotherapy and ART.

The eight patients who were treated with paclitaxel due to treatment failure included five with disseminated skin/oral or visceral disease (Lilongwe stage 4 disease) and three with persistent symptoms of woody edema (Lilongwe stage 3 disease) that impacted patient’s quality of life. Of these eight patients, seven had been previously treated with ABV and one had previously been treated with BV. Outcomes stratified by reason for paclitaxel are presented in Table 2. Of the five patients with visceral/disseminated disease, three achieved CCR and 2 achieved PR. (One of the patients who achieved CCR died of non-KS related causes four years after completion of paclitaxel). Of the three patients with woody edema, one patient achieved PR, one patient was alive but had active disease at study censure and one patient died of complications of severe acute malnutrition five days after initiating paclitaxel.

The five patients who received paclitaxel as initial treatment included two with lymphadenopathic disease (Lilongwe stage 2) and three with disseminated skin/oral or visceral disease (Lilongwe stage 4 disease). Paclitaxel was selected as initial treatment for the following reasons: two patients had experienced anaphylaxis with BV or ABV, one patient had visceral/disseminated disease and was given paclitaxel due to clinician preference, one patient was receiving treatment for multidrug resistant tuberculosis when diagnosed with KS and paclitaxel was selected to minimize the risk of drug–drug interactions; and one patient had chronic lung disease induced cardiomyopathy requiring an anthracycline sparing chemotherapy regimen.

The four patients who were treated with paclitaxel due to KS relapse included three with lymphadenopathic disease (Lilongwe stage 2) and one with woody edema (Lilongwe stage 3). Of these four patients, two had previously been treated with BV and two had previously been treated with ABV. The patient who died had lymphadenopathic KS and died of suspected PCP, severe acute malnutrition, and uncontrolled HIV four months after initiating paclitaxel.

## 4. Discussion

We present an in-depth description of the use of paclitaxel to treat KS in children and AYAs in a LMIC setting which demonstrated that such treatment is both feasible and effective across varying degrees of KS disease. Favorable outcomes were possible despite the challenges involved in treating a clinically complex cohort with varying KS disease severity—70.6% (12/17) had relapsed KS or experienced treatment failure, 70.6% (12/17) had T1 disease, and 47.1% (8/17) had visceral and/or disseminated skin or oral disease. This cohort also had considerable co-morbidities, 62.5% (10/17) had WHO severe immunosuppression, and 35.3% (6/17) had severe acute malnutrition. Moreover, the cohort’s follow up time of 37.3 months suggests that these patient’s responses to paclitaxel were durable and life-sustaining despite the complexity of this cohort.

These results shine a ray of hope across the difficult, rocky landscape of treating pediatric and AYA patients with KS in LMIC settings. We were encouraged that paclitaxel was an effective, life-saving treatment for many patients, including those with KS relapse, and hope other settings with high burdens of pediatric KS will consider its uptake and use in pediatric and AYA treatment regimens. In a cohort description of pediatric KS patients from Malawi, patients with visceral and/or disseminated skin/oral KS often experienced events and did not achieve CCR with BV treatment alone [20]. However, in this cohort, while 62.5% (5/8) of patients with visceral and/or disseminated skin/oral KS had experienced treatment failure, with paclitaxel treatment, 62.5% (5/8) achieved CCR, and 87.5% (7/8) were alive, mirroring experiences in Mozambique and Botswana in which even pediatric patients with severe KS disease were able to achieve positive outcomes [16,17]. Patients who received paclitaxel as initial treatment trended towards lower mortality rates and higher CCR/PR rates compared to those who received paclitaxel for relapse or treatment failure, but these differences were not statistically significant due to small sample sizes.

In this cohort, four patients presented with features of KS-IRIS at the time of KS diagnosis. Pediatric KS cohort descriptions have reported varying amounts of KS IRIS—cohort descriptions from Malawi have reported rates of 22% [28] and 25.7% [20]. In this cohort, patients who presented with features of KS IRIS did not require adjunctive treatment for IRIS and improved with chemotherapy treatment; one was treated with BV, one with ABV, and two received initial treatment with paclitaxel.

The patient who was HIV negative with endemic KS presented with lymphadenopathy and skin lesions which were histopathologically confirmed to be KS. She was initially treated with ABV and achieved CCR; however, she relapsed five months after completing treatment with ABV. With paclitaxel treatment, she achieved complete clinical remission. Her case highlights the important therapeutic role paclitaxel can play in the management of pediatric patients with endemic KS and the importance of ensuring patients with endemic KS can access paclitaxel treatment [29].

This cohort also experienced a reassuring safety profile with paclitaxel, as no patients experienced neurotoxicity or paclitaxel-induced anaphylaxis or severe allergic reactions, and other side effects were seen infrequently. The favorable safety profile seen in this pediatric cohort reassuringly was in line with other pediatric case reports that safely used paclitaxel to treat KS in low- [16], middle- [30], and high-income [31] settings, and to treat other pediatric cancers [32,33]. Similarly, in a large randomized controlled trial of adults with HIV associated KS in LMIC, paclitaxel was well tolerated with a favorable safety profile [15]. With quality of life being a cornerstone of all pediatric palliative care, these positive safety findings for paclitaxel are crucially important and help bolster support for its use across a variety of settings. While it is true that many LMIC lack the sophisticated diagnostics needed to monitor for all potential drug toxicities, these reassuring case reports continue to make a strong argument that the life-affirming benefits of paclitaxel in children and AYA with severe KS outweigh potential toxicities.

A consideration of the costs of chemotherapy is of particular importance in any LMIC setting. Analyzing the known local costs of chemotherapy at this site in southwestern Tanzania, the average six-month chemotherapy costs (in USD for a patient with BSA of 1.0 m^2^) were the following: Paclitaxel: $349, BV regimen: $183, and ABV regimen: $260. Thus, a paclitaxel-only regimen is around twice as expensive as a BV regimen and 1.3× more expensive than ABV. While these figures only consider the cost of the chemotherapy agents, there is a strong argument that the modest 1.3–1.9× cost increase of using paclitaxel would be well justified given its effectiveness and favorable toxicity profile, both of which would help reduce future downstream costs related to lab monitoring, chemotherapy toxicity management, relapse/treatment failure, and additional clinic visits and/or hospitalizations.

As a small retrospective cohort analysis, this study has several limitations. Due to resource limitations, histopathologic confirmation of KS diagnosis was not possible for many patients and chemotherapy toxicity monitoring was largely clinical. Additionally, the sample size was small and from a single site which limited statistical evaluation.

The existing pediatric literature is limited to small descriptions of outcomes in children with KS after treatment with paclitaxel [16,17], and while randomized controlled trials in adults with KS in LMIC have demonstrated paclitaxel to be superior to other chemotherapy regimens [15], no such comparisons exist in children and adolescents with KS. Future studies could evaluate the use of paclitaxel as first line therapy for children and adolescents with KS in a randomized trial with other more commonly utilized first line therapies such as BV or ABV, explore patient and caregiver satisfaction with its use, and seek to determine which pediatric and adolescent patients with KS could most benefit from paclitaxel.

## 5. Conclusions

In conclusion, this analysis demonstrated that treating children and AYA with KS with paclitaxel was effective, safe, and feasible. The majority of patients achieved favorable clinical outcomes, as 82.4% (14/17) of patients survived and no patients were lost to follow up, despite the severity of these patient’s illness and the resource limitations of the setting where they were treated. We hope these findings can be used to advocate for an increased role of paclitaxel in children and AYA who are living with difficult to treat KS worldwide.

## Figures and Tables

**Figure 1 children-08-00275-f001:**
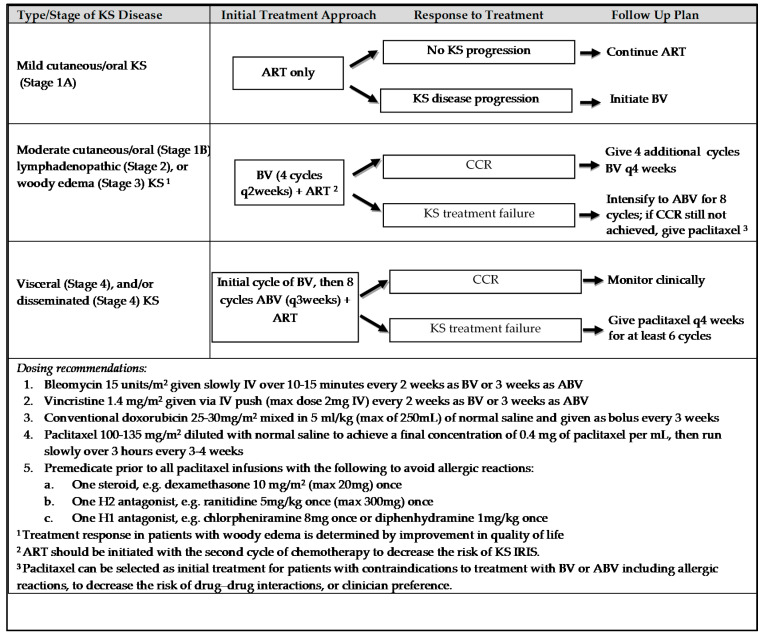
Treatment algorithm for pediatric and adolescent patients with Kaposi sarcoma. Adapted from the risk-stratified and response-adapted therapeutic approach to pediatric Kaposi sarcoma [18] and recommended chemotherapy plan for KS in pregnancy in resource limited settings [21]; Abbreviations: ART—antiretroviral therapy; KS—Kaposi sarcoma; BV—bleomycin + vincristine; ABV—doxorubicin + bleomycin + vincristine; CCR—complete clinical remission; IRIS—immune reconstitution inflammatory syndrome; IV—intravenous.

**Table 1 children-08-00275-t001:** Demographic and clinical characteristics of children and adolescents, and young adults who were treated with paclitaxel at time of KS diagnosis (*n* = 17).

Characteristic at Time of KS Diagnosis	Result
Age in Years (Median, IQR)	13.0 (9.3–15.8)
Children (0–10 years)	5 (29.4%)
Adolescents (10–19 years)	11 (64.7%)
Young adults (20–25 years)	1 (5.9%)
Male sex	10 (58.8%)
Diagnosis supported by histopathology	3 (17.6%)
KS Clinical Features	
Hyperpigmented Skin Lesions	14 (82.4%)
Lymphadenopathy	13 (76.5%)
Subcutaneous Nodules	10 (58.8%)
Woody Edema	9 (52.9%)
Oral lesions	10 (58.8%)
Clinical GI involvement	3 (17.6%)
Clinical Pulmonary involvement	5 (29.4%)
Disseminated disease (>20 skin lesions)	7 (41.2%)
Lilongwe Stage	
Stage 1 (Mild/Moderate KS limited to skin/oral involvement)	0 (0%)
Stage 2 (Lymphadenopathic KS)	5 (29.4%)
Stage 3 (Woody Edema KS)	4 (23.5%)
Stage 4 (Visceral and/or Disseminated skin/oral KS)	8 (47.1%)
TIS Stage ^1^	
T0	5 (29.4%)
T1	12 (70.6%)
I0 ^2^	6 (37.5%)
I1	10 (62.5%)
S0	5 (29.4%)
S1	12 (70.6%)
Cytopenias	
Severe anemia (Hemoglobin < 8 g/dL)	2 (11.8%)
Severe thrombocytopenia (Platelets < 50,000/mm^3^)	2 (11.8%)
Severe Acute Malnutrition	6 (35.3%)
Moderate Acute Malnutrition	4 (23.5%)
HIV+	16 (94.1%)
WHO severe immunosuppression	10 (62.5%)
WHO advanced immunosuppression	2 (12.5%)
WHO mild immunosuppression	1 (6.2%)
No immunosuppression	3 (18.8%)
On ART at diagnosis	14 (87.5%)
Median time on ART prior to KS diagnosis	5.3 months (IQR 1.5–93.8)
On ART < 6 months	7 (50%)
On ART > 6 months	7 (50%)
On NNRTI based regimen prior to diagnosis	12 (85.7%)
On PI based regimen prior to diagnosis	2 (14.2%)
VL obtained (for those on ART > 6 months at diagnosis)	7 (n = 7)
VL < 1000 cp/mL	2 (37.5%)
VL > 1000 cp/mL	5 (62.5%)
IRIS	4 (25%)

^1^ T0 refers to disease limited to the skin and/or lymph nodes and/or minimal oral disease; T1 refers to tumor associated edema, nodular palate lesions and/or visceral disease. I0 refers to patients with CD4 > 200 cells/µL or >15% if < 5 years of age; I1 refers to patients with CD4 < 200 cells/µL or <15% if < 5 years of age; S0 refers to patients without systemic illness; S1 refers to patients with systemic illness. ^2^ 1 patient was HIV negative; Abbreviations—ART: antiretroviral therapy; IRIS: immune reconstitution inflammatory syndrome; IQR: interquartile range; GI: gastrointestinal; KS: Kaposi sarcoma; NNRTI: non-nucleoside reverse transcriptase inhibitor; PI: protease inhibitor; VL: viral load: WHO: World Health Organization.

**Table 2 children-08-00275-t002:** Clinical outcomes of pediatric, and adolescents and young adults (AYA) with KS stratified by reason/context for receiving paclitaxel.

Clinical Characteristic	Upfront Paclitaxel (*n* = 5)	Relapse KS (*n* = 4)	Treatment Failure (*n* = 8)	No Paclitaxel (*n* = 54)	*p*-Value
Alive	100% (5/5)	75% (3/4)	75% (6/8)	70% (38/54)	0.64
Died	0% (0/5)	25% (1/4)	25% (2/8)	30% (16/54)	0.64
CCR	100% (5/5)	75% (3/4)	38% (3/8) ^‡^	54% (29/54)	0.08
PR	0% (0/5)	0% (0/4)	38% (3/8)	17% (9/54)	0.35

^‡^ One patient with treatment failure KS achieved CCR but died of non-KS related causes after completing treatment. Abbreviations—CCR: Complete clinical remission; KS: Kaposi sarcoma; PR: partial response.

## Data Availability

The datasets used and/or analyzed during the current study are available from the corresponding author on reasonable request.

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
