# Peer review of "Use of Paclitaxel to Successfully Treat Children, Adolescents, and Young Adults with Kaposi Sarcoma in Southwestern Tanzania"

_children, 2021, doi:10.3390/children8040275_

Round 1

Reviewer 1 Report

This is a well written and important contribution to the literature. Very few pediatric cohort KS studies have been published and fewer in SSA with paclitaxel as a treatment option.

The paper could be strengthened by further inclusion of the 71 patients who were treated for KS and did not require/receive paclitaxel…how did the 54 who did not go on the paclitaxel pathway do? Describing the whole cohort would strengthen the paper. This seems a major edit (a strong suggestion, but optional).  The focus on the paper could remain on paclitaxel, but incorporate the other 54 participants into Table 1. For example, keep N-17 for paclitaxel, but show the other 54 and their treatment with BV or ABV or nothing (just ART?) and their characteristics as a column next to that presented for paclitaxel. In Table 2 could also show the outcomes Alive/Dead, CCR, and perhaps add PR as an outcome for completeness and incorporate space for the 54 who did not receive paclitaxel.

Additional comments....

Abstract:

AYA only needs to be defined once and then use the AYA abbreviation. (the same comment applies to line 69 in the intro and throughout the manuscript)

Are AYA considered “pediatric”. If yes, you can just say a retrospective pediatric cohort in the second sentence. (same for line 231 and other parts of the manuscript)

The next sentence paclitaxel was given also as first line to 5 patients. In this case, it should be noted in "special circumstances" could be initial therapy.

Intro: read fine. no comments

Methods: A treatment algorithm would be helpful, perhaps as a figure 1.

As in the abstract, paclitaxel was not mentioned as initial treatment, but we learn N=5 did receive it as initial therapy. Including this as a pathway option for special circumstances (allergic reasons/clinican preference/reduce DD interactions, etc.) is important to mention. You could apply the staging system in the figure as well.

Explanation of the staging system is useful (this is somewhat defined in Table 1), but a brief rationale as to why you created/use the Lilongwe staging (why this instead of TIS) and why you retrospectively apply TIS will also be helpful. An example of what is the difference between T0 and Lilongwe stage 1? Is that the lymph node disease (Lilongwe stage 2) 5 = T0. In otherwords, you break up T0 in to stage 1 and 2 and T1 into stage 3 and 4?

For I in the TIS system, what cutoff are you using as this went from <200, <150, and later <100. How does this compare with the WHO definitions in your table 1. Please define here in the methods, don’t ask your audience to look it all up themselves!

Define severe v moderate acute malnutrition

Results: Would be great to include info on all 71 before you focus solely on the 17 who received paclitaxel. How many were new HIV diagnosis? How many progressed from BV to ABV intensification due to partial response?

Did any develop IRIS? How did you manage it?

It is confusing to go from the TIS staging system to the Lilongwe staging system without explaining the differences in the methods.  Table 1 helps clarify, but see suggestions in the methods.

My understanding is T0 disease is confined to the skin/lymphnodes T1 is more extensive (visceral, extensive oral lesions, ulcerations/edema, etc.). Not knowing what your stage 4 is it is unclear what 70% T1 v. 47% L.stage 4 disease means until the table, and essentially your stage 3 + 4 = T1 and your stage 1 + 2 = T0? Is that correct?? Just spelling that out in the methods (if correct) will be helpful.

Table 1: for VL for those receiving ART >=6 months N=6, but for VL (art>=6mo) N=8? Why are these different? For those <1000 were they not detected or still above 200 or 400?

Lines 166-192 are neatly summarized in Table 2 and perhaps less detail is needed in the text, rather reference Table 2.

Lines 193-197 are perfect and important, but should probably be moved to the discussion as it is your interpretation of the results.

Table 2: AYA is misspelled in the title.

I like the *footnote explanation, but you only need this explanation in one place…either text or table and only need abbreviations for things used in the table if you leave the explanation in the text.

Discussion: Typically refrain from personal pronouns such as “our” study and “our” cohort. Better to say this study, the study cohort, etc.

What was the underlying dx of the one patient who was HIV neg? Why was this patient so immunosuppressed? Are you calling their KS endemic? Is pathophysiology for this case similar enough to those with epidemic or HIV associated KS to keep them in the cohort?

The discussion would benefit from mention of any IRIS-KS cases in the cohort.

Also a bit of cost comparison/cost effectiveness would be helpful. What was the cost of paclitaxel and who paid for the treatment? Pegylated liposomal doxorubicin (PLD) is another promising agent….any thoughts on use in pediatrics? What is the cost comparison between PLD and paclitaxel and v. BV/ABV/etoposide?  If $$$$ can you make a compelling case for it to become stnd/1st line tx in Africa?

Reviewer 2 Report

The Authors report a retrospective study of a cohort of young patients with Kaposi Sarcoma treated with paclitaxel, reporting the safety and efficacy of this drug. 

The main limit of the study is that it is retrospective. Another big limit is the way to diagnose the tumor that is histological only in a minority of cases. Even the follow up of patients concerning the toxicity of treatment is not really correct, due to the low resources of the Country. Finally the number of patients is small. Anyway the Authors really underline these limits.

Apart form this, the study is well conducted and results are interesting.

I suggest not to use the percentage in the results, considering the small number of patients.

Line 101: the definition of PR is really strange. How do you measure the disease, for example in the gastrointestinal form?

In table 2 AYA, not ALA

251: a randomized trial with usual and experimental treatment should be more interesting.
